# Arterial Blood Supply to the Cerebral Arterial Circle in the Selected Species of Carnivora Order from Poland

**DOI:** 10.3390/ani13193144

**Published:** 2023-10-08

**Authors:** Maciej Zdun, Jakub Jędrzej Ruszkowski, Aleksander F. Butkiewicz, Maciej Gogulski

**Affiliations:** 1Department of Animal Anatomy, Poznan University of Life Sciences, Wojska Polskiego 71C, 60-625 Poznan, Poland; maciejzdun@umk.pl; 2Department of Basic and Preclinical Sciences, Nicolaus Copernicus University in Torun, Lwowska 1, 87-100 Torun, Poland; 304242@stud.umk.pl; 3University Centre for Veterinary Medicine, Szydłowska 43, 60-656 Poznan, Poland; maciej.gogulski@up.poznan.pl; 4Department of Preclinical Sciences and Infectious Diseases, Poznan University of Life Sciences, Wołynska 35, 60-637 Poznan, Poland

**Keywords:** angiology, circle of Willis, anatomy

## Abstract

**Simple Summary:**

Carnivores are a wide, diverse group of mammals whose representatives live all over the world. The study describes arterial blood supply to the cerebral arterial circle of the group of selected species in the Caniformia suborder living in Poland. The results were discussed based on the current knowledge of this field of research.

**Abstract:**

Carnivores are a wide, diverse group of mammals whose representatives live all over the world. The study presents the results of the analysis of the arterial vascularization of the blood supply to the cerebral arterial circle of selected species in the Caniformia suborder living in Poland. The selected group consists of wild and farm animals—105 animals in total. Three different methods were used—latex preparation, corrosion cast, and cone-beam computed tomography angiography. The main source of blood for encephalon in the described species is the internal carotid artery, and the second one is the vertebral artery. The results were discussed in relation to the current knowledge of this field of research. Information on the potential physiological meaning of such vascular pattern has been provided.

## 1. Introduction

Carnivores (order Carnivora) are a diverse group of mammals that managed to populate many different habitats on all of the continents [1]. According to the International Union for the Conservation of Nature, there are 290 species belonging to this order [2]. The order consists of two suborders—Feliformia and Caniformia. In the study, only representatives of the Caniformia suborder were used. The described species belong to the Polish fauna and are members of four families—Canidae, Mustelidae, Procyonidae, Phocidae. Red fox (*Vulpes vulpes*), gray wolf (*Canis lupus*), European badger (*Meles meles*), Eurasian otter (*Lutra lutra*), and gray seal (*Halichoerus grypus*) are native to Polish fauna. The raccoon dog (*Nyctereutes procyonoides*), American mink (*Mustela vison*) and common raccoon (*Procyon lotor*) are invasive species. American mink is also a fur animal that is kept on farms.

The vascular patterns of arteries of different animal species have been the subject of anatomical research for decades. Various, often species-specific, vascular systems have been described in many species of carnivorous animals, often focusing on specific parts of the body. The cerebral arterial circle is among the most frequently described anatomical regions in this aspect [3,4,5]. The blood supply to these structures is often overlooked. The aim of this study was to assess and describe detailed arterial patterns of the arteries supplying blood to the arterial circle of the brain in the described species.

## 2. Materials and Methods

### 2.1. Animals

The study was conducted on 105 specimens of the Carnivora order of 8 species (Table 1). The animals used were adults of both sexes. Only animals without the trauma of the head and neck region were included in the research group. The animals were obtained from hunters, breeders, and zoos. All animals were obtained as post-mortem material. No animals were killed for the purpose of the study. The species distribution and number of individuals used in the study are presented in Table 1.

### 2.2. Methods

In the study, different anatomical methods were used to obtain a high-quality, complete image of the vascular pattern of the described area.

The classical anatomical preparation methods used in the study included latex preparation (method 1) and corrosion cast (method 2). A more advanced imaging method was the use of maximum-intensity projection reconstruction of cone-bean computed tomography scans (method 3). While working with the cadavers, additional precautions were taken. Researchers were wearing masks with high-quality filters and were working in a preparation room with an efficient ventilation system. The system settings were 20 air changes per 1 h.

Method 1

This method was used in 28 specimens. The method consists of injecting bilateral common carotid arteries with liquid, red LBS 3060 latex. After the injection, the preparations were cured in 5% formaldehyde solution for 14 days. The next step was rinsing specimens with running water for 48 h to flush out the excess formaldehyde. The next stage consisted of the manual dissection of soft tissues. The excess connective tissue was cut, which resulted in red arteries being obtained from the surrounding soft tissues.

Method 2

This method was used in 67 preparations. The method consists of injecting bilateral common carotid arteries with a tinged solution of the chemo-setting acrylic material Duracryl^®^ Plus (SpofaDental, Jičín, Czech Republic). This material hardened after the injection, and the specimens were submerged in the detergent solution (Persil, Düsseldorf, Germany) for the process of maceration. The temperature of water used for this process was 42 °C. The process lasted 28 days. This method produced a red acrylic cast of the arterial vessels on the bone scaffold.

Method 3

This method was used in 10 preparations. The method consists of administering contrast agent (barium sulphate; barium sulphuricum 1.0 g/mL, Medana, Sieradz, Poland) to bilateral common carotid arteries. The scans were performed at the University Centre for Veterinary Medicine in Poznan, Poland, with the use of Animage Fidex computed tomography (Fidex Animage, Pleasanton, CA, USA).

After the examination, the scans were studied, and proper images were taken in FidexGUI (version 3.6.0, Animage, USA) with maximum-intensity projection image reconstruction.

The names of the anatomical structures were standardized according to Nomina Anatomia Veterinaria [6].

All of the photographs taken during the study were taken with a digital camera (Canon EOS 250D). The photographs were saved in JPG format. GIMP v2.10.18 digital image editing software was used to process the photographs.

## 3. Results

The main source of blood for encephalon is the internal carotid artery (arteria carotis interna). This artery branches off from the common carotid artery (arteria carotis communis) at the point where the main arterial stream becomes the external carotid artery (arteria carotis externa). The internal carotid artery at the initial segment creates a thickening called the carotid sinus (sinus caroticus) (Figure 1, Figure 2, Figure 3 and Figure 4). This is most demonstrable in the gray seal, next in the gray wolf and red fox, but in six red foxes, its expression is weaker. In the European badger, it is slightly embossed. In the raccoon dog and common raccoon, it is poorly marked and takes on a more elongated, less convex shape. In the American mink and Eurasian otter, this vessel branches off with the occipital artery (arteria occipitalis) via a common trunk. This trunk is short and there is no carotid sinus observed. No thickening is observed after the trunk has split into the internal carotid and occipital arteries.

Next, the internal carotid artery heads dorsorostrally and penetrates the carotid canal (canalis caroticus) through the caudal foramen of the carotid canal. In Canidae, this foramen is located near the caudal end of the tympanic bulla (bulla tympanica) (Figure 2 and Figure 5), in the Mustelidae (Figure 6) and common raccoon it is near to the middle of the length of the tympanic bulla, and in Eurasian otter (Figure 7 and Figure 8) it is positioned even more rostrally, in the one-third rostral part of the tympanic bulla. Thus, in this species of carnivores, the internal carotid artery enters the skull more rostrally, and the carotid canal is shorter.

The diameter of the extracranial segment of the internal carotid artery is larger than that of the occipital artery in foxes, European badgers, gray seal and American mink. In the raccoon dog, Eurasian otter and wolf, these vessels are of equal diameter. In the vascular variations, the diameter of the internal carotid artery is smaller than that of the occipital artery in one wolf, while in one Eurasian otter, the internal carotid artery is an artery with a larger lumen. The carotid canal runs along the medial surface of the eardrum. In the carotid canal, the vessel runs rostrally. At the level of the rostral foramen of the carotid canal, it forms a vascular loop, and changes direction by 180° (Figure 2 and Figure 5). For a short distance, it runs caudally. Then, it circles an arc once again, this time with a more gentle course, and heads dorsally, entering the cranial cavity. This vascular loop at the level of the rostral foramen of the carotid canal protrudes from the foramen in the gray wolf. This vessel protrudes slightly or is on the border of the foramen in the fox, raccoon dog and Eurasian otter; it does not protrude in the European badger. In the American mink, common raccoon and gray seal, this artery does not create a vascular loop. In the American mink via the rostral foramen of the carotid canal enters the branch from the ascendance pharyngeal artery (arteria pharyngea ascendens) and joins the internal carotid artery. In the fox, gray wolf, European badger and raccoon dog this branch from the ascendance pharyngeal artery joins the vascular loop (Figure 5). In other species, no connection was observed between the branch of the ascending pharyngeal artery and the internal carotid artery, although this vessel ran in close proximity to the aforementioned foramen. In the Eurasian otter, from the internal carotid artery branched off the small vessel that heads to the caudal wall of the pharynx. No ascending pharyngeal artery from the external carotid artery was observed. Before the internal carotid artery begins to form the cerebral arterial circle, it is joined by an anastomosing branch from the external ophthalmic artery (ramus anastomoticus) (Figure 9). The external ophthalmic artery is a branch from the maxillary artery (arteria maxillaris). This last vessel is a continuation of the external carotid artery. In the common raccoon, this connection is observed between the external ethmoid artery (arteria ethmoidalis externa) and the internal carotid artery. In two specimens, this connection is between the external ophthalmic artery and the internal carotid artery. No such connection was observed in the Eurasian otter, American mink and gray seal. The internal carotid artery then divides into the rostral cerebral artery (arteria cerebri rostralis) and the caudal communicating artery (arteria communicans caudalis). The latter is joined to the basilar artery (arteria basilaris).

The third source of blood is the vertebral artery (arteria vertebralis) (Figure 10). This vessel passes into the transverse process foramen (foramen processus transversus) of the cervical vertebrae.

Between the second and third cervical vertebra branches off the medial branch of the vertebral artery. Bilateral branches form the ventral spinal artery (arteria spinalis ventralis) (Figure 11). This artery heads cranially and joins the basilar artery. Moreover, the vertebral artery heads cranially and enters the transverse process foramen of the atlas. Then, it exits more cranially under the wing of the atlas. At that point, the anastomosing branch to the occipital artery (ramus anastomoticus cum a. occipitali) branches off. Such arterial connection was not observed in a seal. Next, the vertebral artery enters the lateral vertebral foramen (foramen vertebrale leterale) of the atlas, penetrates the vertebral foramen (foramen vertebrale) and joins the basilar artery. This pattern is present in the Eurasian otter, American mink, foxes and European badger. In the common raccoon, no ventral spinal artery joining the basilar artery was observed. In the gray seal, no anastomosing branch to the occipital artery was observed. Moreover, in this species, the ventral spinal artery branched off between the first and second cervical vertebra.

## 4. Discussion

The well-developed internal carotid artery is the main source of blood for the cerebral arterial circle in the dog [7,8,9], as well as the Arctic fox [7,10], common fox [11,12] silver fox [13], raccoon dog, and species of the seal family (Phocidae), mustelids family (Mustelidae), bear family (Ursidae), raccoon family (Procyonidae) [7] of the order Carnivora. In Feliformia, the second group of Carnivores, the extracranial part of the internal carotid artery is not present in adult animals [14,15]. In fetuses and young cats, this artery provides blood to the encephalon, but at about 4–8 weeks of age this artery is incomplete and the connection between the common carotid artery and the cerebral arterial circle ceases to function [15]. The connection of the cerebral arterial circle to the internal carotid artery has been found in some rodents (Rodentia): in the European beaver [16] and Canadian beaver [17], the Egyptian spiny mouse [18], the American muskrat [7], the rat [7,19] and the ursine [7]. A fully preserved internal carotid artery, which is the main source of blood to the brain, is found in all representatives of the odd-toed ungulates, i.e., the horse and other representatives of the family Equidae [7,20,21], in tapirs of the family Tapiridae and the rhinoceros of the family Rhinocerotidae [7]. It is also found in the rabbit and hare of the order Lagomorpha [7,22], in the Abyssinian highlander of the order Hyracoidea, in the wallaby and red kangaroo of the order Marsupialia, the two-toed sloth of the order Xenarthra, in the primates of the order Primates [7,23,24], and in the elephant of the order Proboscidea [25,26]. The different courses of this vessel in various animal species were described. In the horse, the course of the internal carotid artery is straight and it does not form a bend before entering the cranial cavity. This vessel runs on the dorsal and rostral surface of the medial compartment of the guttural pouch and passes through the ragged opening—foramen lacerum [27,28]. Then, it enters the cranial cavity, where it passes through the ventral petrosal sinus and enters the venous cavernous sinus; here, it forms an S-shaped curve [29,30]. Such a course also occurs in the donkey [31]. Similarly, in the dog, the artery takes a fairly direct course by way of the jugular foramen, through the occipito-tympanic fissure and into the cavernous sinus [32]. Thus, this vessel does not pass between the eardrum and scalene parts of the temporal bone, and thus does not interfere with sound perception, as pointed out by Zedenov [33]. A preserved internal carotid artery is also found in dolphins or narwhals. However, this vessel extends into the tympanic cavity, passing through the middle ear in a semicircular arc [34,35]. As is well known, for these marine mammals, their sense of hearing is their most important sense, and they use it to emit and receive infrasound. In light of this information, it can be assumed that the obliteration of this vessel in ruminants is not related to their emission of low-frequency sounds, but is merely the result of developmental changes associated with a change in the position of the eardrum portion of the temporal bone. To be sure, it would be necessary to compare the frequency range of the waves emitted by the vessel with the range of perceived sounds in these aquatic mammals, but Zedenov [33] does not specify such a range. In animals in which obliteration does not occur, the course of this vessel is different and the development of the cranial skeleton does not affect the course of the internal carotid artery. Strategies of vascularization of the encephalon are different. In some rodents, the basilar artery is the main source of blood. In the guinea pig, it directs as much as about 66% of the blood to the cerebral arterial circle [36]. A strong basilar artery has furthermore been described in the aguti [37], the porcupine [38], the capybara [39], the common degu [40], nutria [41], European ground squirrel [42], chinchilla [43] or red squirrel [44]. In camels, the basilar artery has a relatively large lumen; however, it does not supply this organ to such an extent [45]. In the dromedary, the internal carotid artery is responsible for supplying 13% of the blood to the rostral epidural rete mirabile [46], and the vessels emerging from the rete mirabile are primarily responsible for cerebral vascularization. In the dog, the internal carotid artery is also a strong vessel, accounting for half the diameter of the external carotid artery [32]. In ruminants, despite the obliteration of the internal carotid artery, the basilar artery does not contribute significantly to the blood supply to the brain, with the maxillary artery being of greatest importance [47,48,49]. In addition to the obvious role of this vessel in supplying blood to the brain, other roles of this vessel have also been considered. Maloney et al. [50] sought to test the hypothesis of the role of the internal carotid artery in selective cooling of the brain in the horse. This hypothesis relied on the exchange of heat between the blood in the vessel and the air sac with which it comes into contact. However, the authors themselves stated that, assuming that the contact between these structures is about 6% of the surface area of the bag, it is not possible for the air to flow through the air sac in such a way that the temperature of the blood in the vessel can be realistically reduced. The phenomenon of selective cooling of the brain has been described in ruminants. However, the rostral epidural rete mirabile is involved, along with the venous cavernous sinus, and the internal carotid artery is not involved. Cooler venous blood returning from the nasal cavity washes over the vessels of the weird network, causing a drop in the temperature of arterial blood flowing into the brain [51,52]. In addition, it is important to note a feature of this artery, which is the serpentine course of the intracranial portion of this vessel. Ruedi [53] equates the probable function of this fragment in the domestic horse to attenuate the pulse wave of arterial blood and protect the brain from a surge of pressure. This course of the vessel was also found in llama [54].

## 5. Conclusions

The results of this study describing blood supply to the arterial cerebral circle in selected species of Caniformia members from Poland enrich the status of the current knowledge in the field of the angiology of the Carnivora order. The results may also contribute as a baseline for further physiological and pathological studies in the field of veterinary medicine.

## Figures and Tables

**Figure 1 animals-13-03144-f001:**
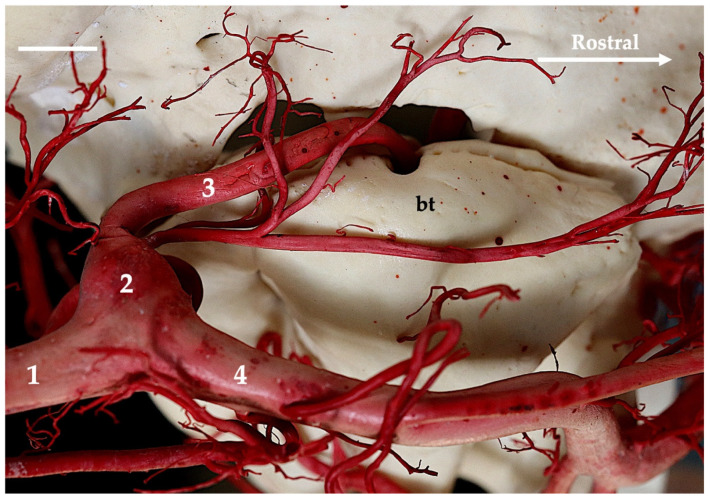
Ventromedial view of the tympanic bulla region of the gray seal. Corrosion cast. The white bar corresponds to a length of 1 cm. bt—tympanic bulla; 1—common carotid artery; 2—carotid sinus; 3—internal carotid artery; 4—external carotid artery.

**Figure 2 animals-13-03144-f002:**
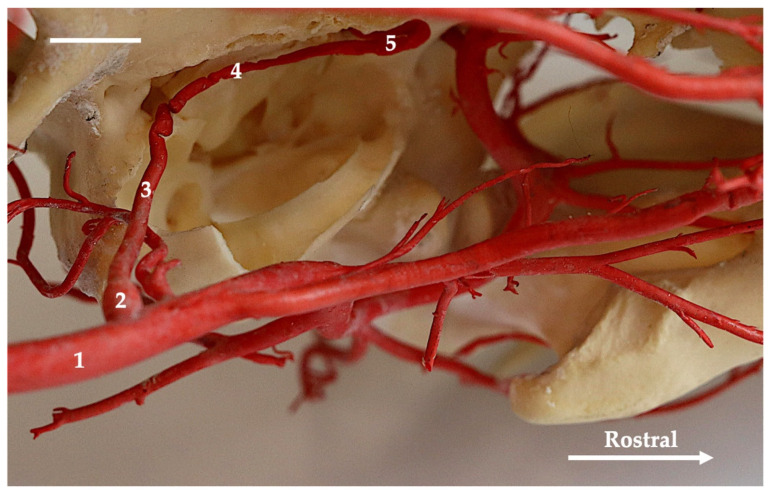
Ventromedial view of the tympanic bulla region of the red fox. Medial part of the tympanic bulla has been removed. Corrosion cast. The white bar corresponds to a length of 1 cm. 1—common carotid artery; 2—carotid sinus; 3—initial part of the internal carotid artery; 4—the part of the carotid artery that penetrates the carotid canal; 5—the vascular look of the internal carotid artery near the rostral opening of the carotid canal.

**Figure 3 animals-13-03144-f003:**
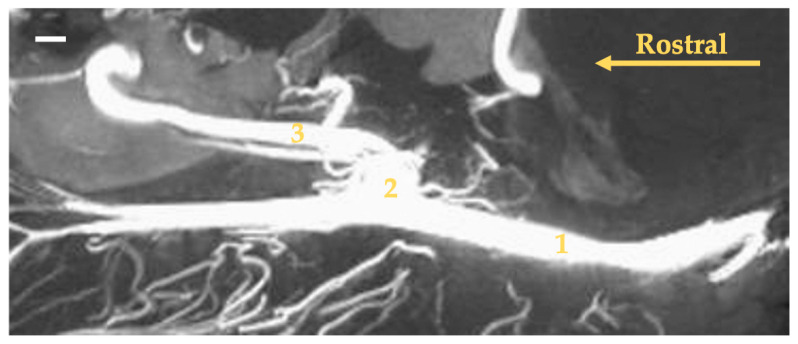
Maximum intensity projection reconstruction of the angioCT scan of the head of the gray seal. The white bar corresponds to a length of 1 cm. 1—common carotid artery; 2—carotid sinus; 3—internal carotid artery.

**Figure 4 animals-13-03144-f004:**
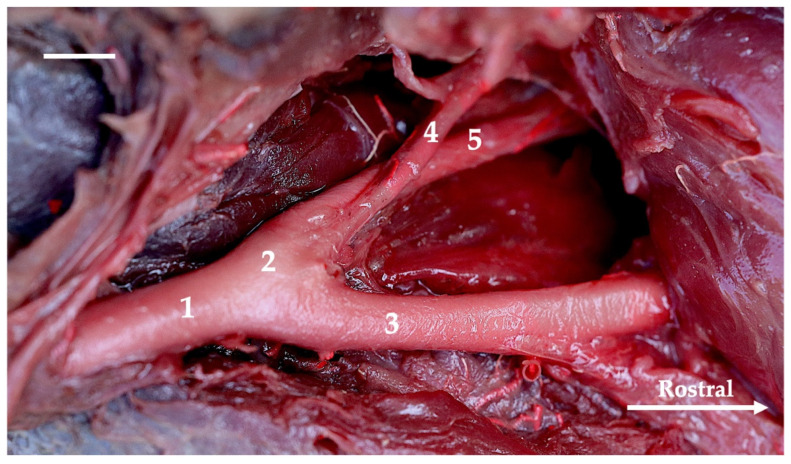
Lateral view of branches of the common carotid artery of the gray seal. Latex preparation. The white bar corresponds to a length of 1 cm. 1—common carotid artery; 2—carotid sinus; 3—external carotid artery; 4—occipital artery; 5—internal carotid artery.

**Figure 5 animals-13-03144-f005:**
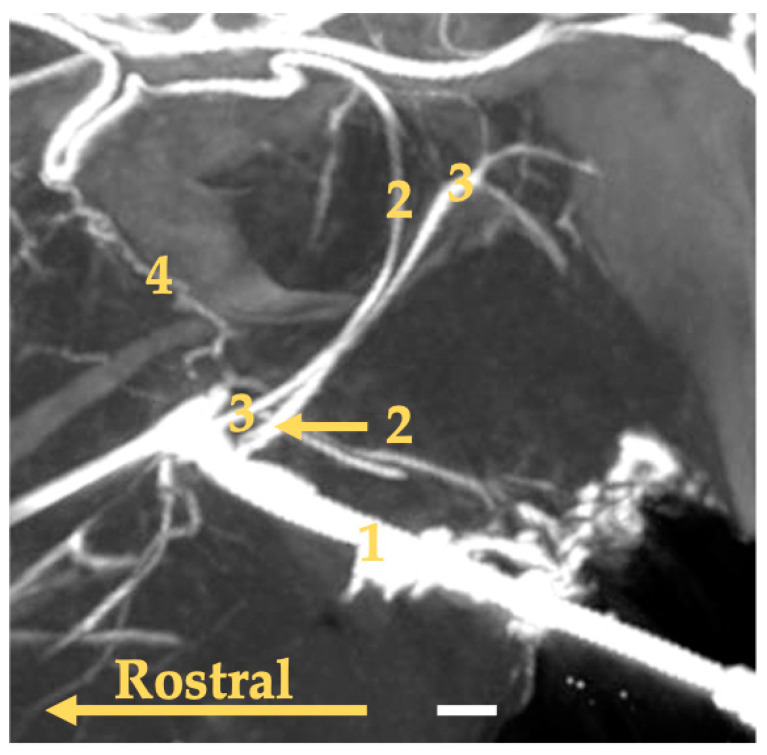
Maximum intensity projection reconstruction of the angioCT scan of the head of the gray wolf. The white bar corresponds to a length of 1 cm. 1—common carotid artery; 2—internal carotid artery; 3—occipital artery; 4—ascending pharyngeal artery.

**Figure 6 animals-13-03144-f006:**
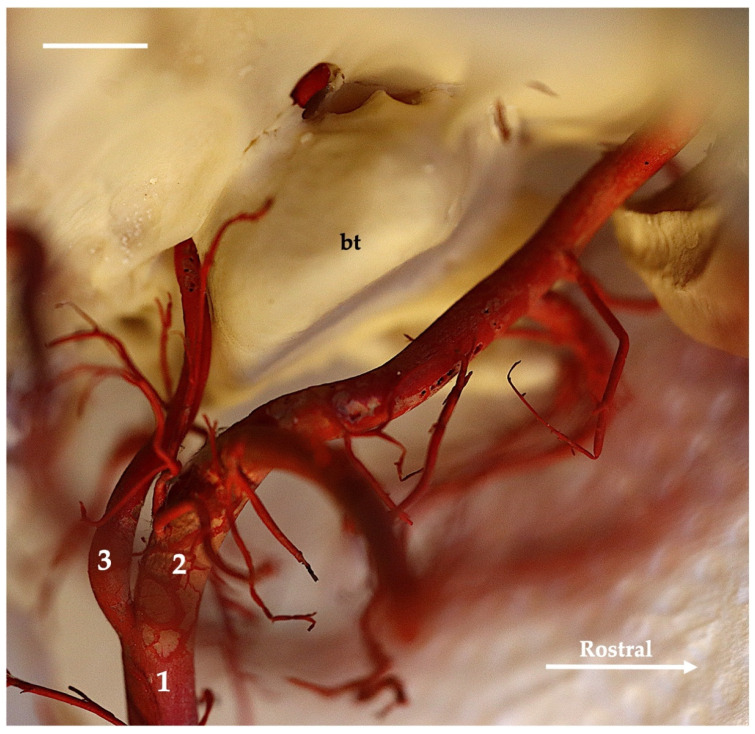
Ventromedial view of the tympanic bulla region of the European badger. Corrosion cast. The white bar corresponds to a length of 1 cm. bt—tympanic bulla; 1—common carotid artery; 2—external carotid artery; 3—internal carotid artery.

**Figure 7 animals-13-03144-f007:**
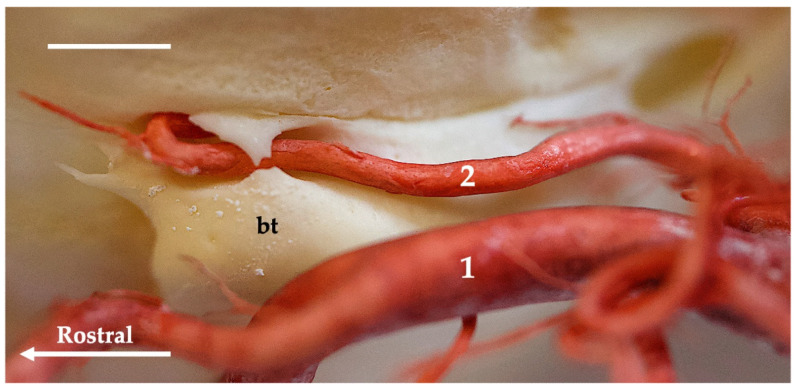
Ventromedial view of the tympanic bulla region of the Eurasian otter. Corrosion cast. The white bar corresponds to a length of 1 cm. bt—tympanic bulla; 1—external carotid artery; 2—internal carotid artery.

**Figure 8 animals-13-03144-f008:**
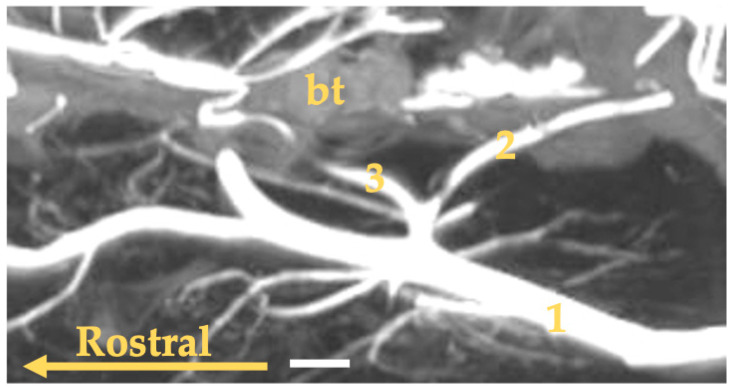
Maximum intensity projection reconstruction of the angioCT scan of the head of the Eurasian otter. The white bar corresponds to a length of 1 cm. bt—tympanic bulla; 1—external carotid artery; 2—occipital artery; 3—internal carotid artery.

**Figure 9 animals-13-03144-f009:**
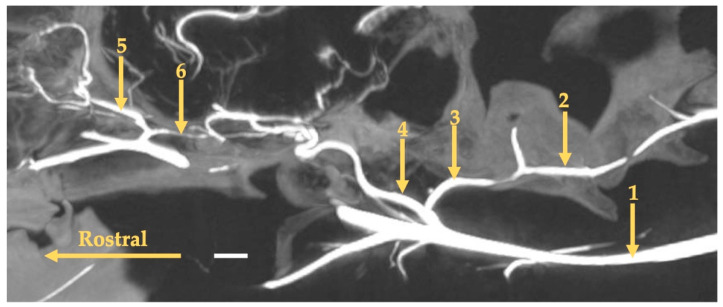
Maximum-intensity projection reconstruction of the angioCT scan of the head of the European badger. The white bar corresponds to a length of 1 cm. 1—external carotid artery; 2—vertebral artery; 3—anastomosing branch to the occipital artery; 4—internal carotid artery; 5—external ophthalmic artery; 6—anastomosing branch from the external ophthalmic artery.

**Figure 10 animals-13-03144-f010:**
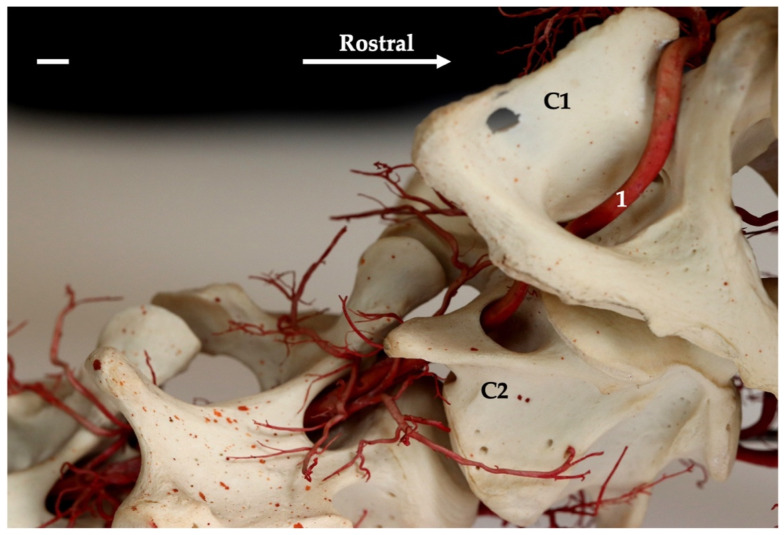
Lateral view of the first three cervical vertebrae of the gray seal. Corrosion cast. The white bar corresponds to a length of 1 cm. C1—first cervical vertebra; C2—second cervical vertebra; 1—vertebral artery.

**Figure 11 animals-13-03144-f011:**
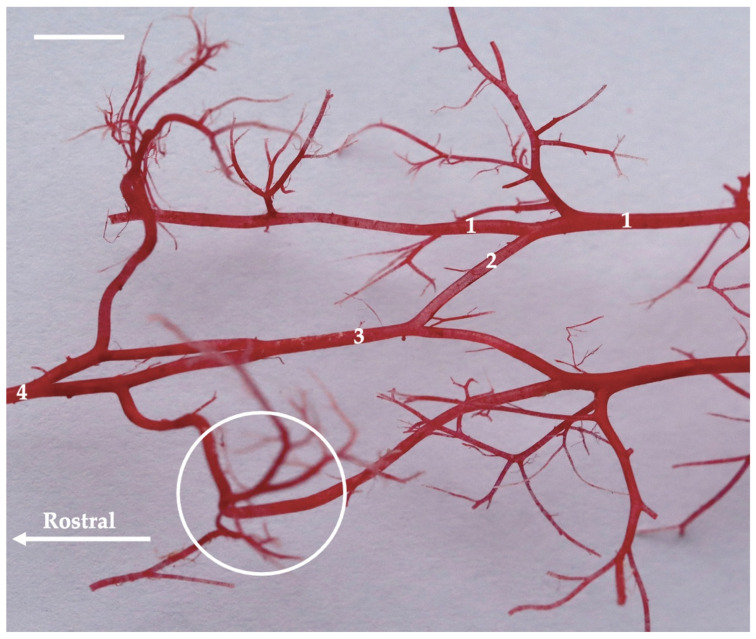
Dorsal view of the arteries of the cervical spine of the American mink. Corrosion cast. The white bar corresponds to a length of 1 cm. 1—vertebral artery; 2—medial branch of the vertebral artery; 3—ventral spinal artery; 4—basilar artery; white circle—the place where vertebral artery wraps around the wing of the first cervical vertebra before entering the vertebral lateral foramen.

**Table 1 animals-13-03144-t001:** The number of specimens examined in this study.

Family	Species	Method 1	Method 2	Method 3
Canidae	Raccoon dog (*Nyctereutes procyonoides*)	5	10	-
	Red fox (*Vulpes vulpes*)	5	13	2
	Gray wolf (*Canis lupus*)	1	2	1
Mustelidae	American mink (*Mustela vison*)	5	13	2
	European badger (*Meles meles*)	5	13	2
	Eurasian otter (*Lutra lutra*)	3	7	2
Procyonidae	Common raccoon (*Procyon lotor*)	3	8	-
Phocidae	Gray seal (*Halichoerus grypus*)	1	1	1

## Data Availability

The data presented in this study are available on request from the corresponding author.

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
