# Peer review of "Arterial Blood Supply to the Cerebral Arterial Circle in the Selected Species of Carnivora Order from Poland"

_animals, 2023, doi:10.3390/ani13193144_

Round 1
Reviewer 1 Report
I suggest inserting calibration bars in the images and replacing image 11. In the literature there are references that classify the arterial circuit. The authors did not use it. Which system predominates? carotid or vertebrobasilar?
Reviewer 2 Report
The work is concerned with the arterial vessels forming the arterial circle of the brain. This is an insufficiently understood subject. The study was performed on carnivorous animals occurring in the area (including invasive species such as raccoon dogs). Poland. As these are animals (except for the wolf and grey seal) that are commonly found, the study groups should include more specimens.
Certain elements of the methodology need to be refined, as they may affect the results obtained. The discussion is broad and includes references to the most important reports in the field. The literature is responsive.
2.2. How the test material (cadavers) was stored before intravascular injection?
2.2. At what time after death and how was the material stored before the CT scan? This is about swelling of the brain tissues interfering with the CT image
188 Change vartebral to vertebral
Reviewer 3 Report
This detailed study is of great importance for the anatomical knowledge and for clinicians.
The images are of good quality depicting the differences among the specimens.
In the discussion some species-specific functional data are taken into account concerning the arterial flux of these arteries that are relevant.
From my point of view, the article should be published as it is a valuable source for anatomists.
Just two considerations:
. In line 231, the authors say that the ICA enters the jugular foramen to reach the carotid canal. They should say the occipito-tympanic fissure. Inside this fissure two openings can be found: the jugular foramen for the IX, X and X cranial nerves and vessels and the caudal opening of the carotid canal. This is a separate entity located more medially inside the occipito-tympanic fissure.
. In the abstract, they state that the anastomotic branch between the ICA and the ophthalmic artery is a source of blood sully to the encephalon. This statement should be proved as it is not reported in any bibliographic source.
Round 2
Reviewer 2 Report
The authors clarified important points about the research methodology. The work may be published.